# Development of a Combined Lipid-Based Nanoparticle Formulation for Enhanced siRNA Delivery to Vascular Endothelial Cells

**DOI:** 10.3390/pharmaceutics14102086

**Published:** 2022-09-29

**Authors:** Yutong He, Dongdong Bi, Josée A. Plantinga, Grietje Molema, Jeroen Bussmann, Jan A. A. M. Kamps

**Affiliations:** 1Laboratory for Endothelial Biomedicine & Vascular Drug Targeting Research, Medical Biology Section, Department of Pathology & Medical Biology, University Medical Center Groningen, University of Groningen, 9713 GZ Groningen, The Netherlands; 2Division of BioTherapeutics, Leiden Academic Centre for Drug Research (LACDR), Leiden University, 2300 RA Leiden, The Netherlands

**Keywords:** lipid-based nanoparticles, cationic lipid formulation, endothelial cell delivery, siRNA therapy, zebrafish model

## Abstract

Low transfection efficiency in endothelial cells (EC) is still a bottleneck for the majority of siRNA-based vascular delivery approaches. In this work, we developed a lipid-based nanoparticle (LNP) formulation based on a combination of a permanently charged cationic lipid-DOTAP and a conditionally ionized cationic lipid-MC3 (DOTAP/MC3) for the enhanced delivery of siRNA into EC. Compared with a single DOTAP or MC3-based benchmark LNP, we demonstrated that the DOTAP/MC3 LNP formulation shows the best transfection efficiency both in primary EC in vitro and in endothelium in zebrafish. The high transfection activity of the DOTAP/MC3 LNP formulation is achieved by a combination of improved endothelial association mediated by DOTAP and MC3-triggered efficient siRNA intracellular release in EC. Furthermore, Ab_VCAM-1_-coupled DOTAP/MC3 LNP-mediated siRNA_RelA_ transfection showed pronounced anti-inflammatory effects in inflammatory-activated primary EC by effectively blocking the NF-κB pathway. In conclusion, the combination of permanent and ionizable cationic lipids in LNP formulation provides an effective endothelial cell delivery of siRNA.

## 1. Introduction

The vascular endothelium is a thin cell layer in all blood vessels, and it functions as a semi-selective barrier between tissues and circulating blood constituents [1]. It also actively mediates many physiological events, such as vascular constriction and relaxation, hemodynamic changes, platelet–leukocyte interactions as well as blood coagulation [2]. Endothelium is a highly specialized tissue, spatially and temporally [3]. This is illustrated by the functional heterogeneity of endothelial cells (EC) in healthy and pathological conditions, which provides possibilities for designing precise targeted therapeutic intervention approaches [4,5]. The endothelium plays a prominent role in the initiation and progression of chronic inflammatory diseases, including but not limited to cardiovascular disease, metabolic disorders, fibrosis, autoimmune diseases and cancer [6,7,8,9,10]. In these diseases, EC become activated, which triggers a cascade of events leading to the expression of cell adhesion molecules, leukocyte recruitment as well as cytokine and chemokine production [11]. The central role of the endothelium in the pathology of inflammatory diseases has prompted interest in endothelium as a therapeutic target for drug delivery purposes [12].

Small interfering RNAs (siRNA) have shown to possess therapeutic value, which is determined by efficient siRNA delivery systems [13,14]. Although various nucleic acid delivery systems aiming at EC have been designed, the low transfection efficiency in EC is still a bottleneck for therapeutic approaches [15,16]. EC, as non-phagocytic cells, generally show inefficient uptake of unmodified nanoparticles and have a limited intrinsic cargo release machinery [17,18]. Therefore, for the endothelial delivery of nucleic acids, attention should be focused on increasing the efficiencies of cellular internalization, processing and cargo release of nanoparticles in EC. Currently, lipid-based nanoparticles (LNP) are considered as one of the most efficient carrier systems for siRNA delivery, and some of them have been recently approved for clinical use [19,20,21]. In terms of transfection efficiency, it has been reported that the design and choice of the cationic lipids in LNP/siRNA systems are decisive for gene silencing in hepatocytes [22]. Both permanent and ionizable cationic lipids have been applied in LNP formulations, and some related studies have been reported, focusing on the investigation of how the cationic lipid composition influences RNA delivery [23,24]. LNP formulated with conditionally ionized cationic lipids with pKa values of 6–7 achieve superior gene silencing due to their ability to facilitate endosomal escape or other intracellular delivery routes [25]. DLin-MC3-DMA (MC3) has been confirmed to be one of the most potent lipids for siRNA delivery formulations, showing an ED_50_ value of 0.005 mg/kg to silence hepatic-specific gene FVII in mice, which was also applied to formulate the first FDA-approved siRNA therapeutic called ONPATTRO™ (Patisiran) for the treatment of transthyretin-mediated amyloidosis [22,26]. On the other hand, LNP containing permanent cationic lipids tend to show improved uptake by target cells, which may be crucial for the delivery of sufficient amounts of siRNA into cells [27]. Given the limited knowledge of how cationic lipid composition influences LNP performance in EC, we explored different cationic lipid-based LNP formulations to develop an advanced LNP/siRNA system for endothelial delivery.

Since permanent cationic lipids improve cellular uptake and ionizable cationic lipids improve intracellular processing, we hypothesized that a combination of permanent and ionizable cationic lipids for siRNA delivery might result in enhanced gene silencing in EC. Three LNP were formulated by changing the cationic lipid composition: (1) LNP with a permanent cationic lipid (DOTAP); (2) LNP with an ionizable cationic lipid (MC3); and (3) LNP with both DOTAP and MC3 in different molar ratios. We examined the in vitro transfection efficiencies of different LNP formulations and the possible mechanisms associated with high transfection efficiency in primary EC. The biodistribution, siRNA intracellular release and gene silencing in vivo were studied in a green fluorescent protein (GFP)-EC transgene zebrafish model. A novel LNP/siRNA formulation for enhanced gene silencing in (activated) EC was developed by comparatively examining cationic lipid excipients as well as optimizing the cationic lipid-based LNP formulation. This formulation is expected to be useful for the treatment of chronic inflammatory diseases by alleviating endothelial activation.

## 2. Materials and Methods

### 2.1. Materials

Lipids dilinoleylmethyl-4-dimethylaminobutyrate (DLin-MC3-DMA or MC3) were obtained from Med Chem Express (Monmouth Junction, NJ, USA). 1,2-Dioleoyl-3-trimethylammonium-propane (DOTAP), 1,2-distearoyl-sn-glycero-3-phosphocholine (DSPC), 1,2-dimyristoyl-rac-glycero-3-methoxypolyethylene glycol-2000 (DMG-PEG), and 1,2-distearoyl-sn-glycero-3-phosphoethanolamine-N-[methoxy (poly-ethylene glycol)-2000]-maleimide (DSPE-PEG-Mal) were obtained from Avanti Polar Lipids (Alabaster, AL, USA). 1,1′-Diooctadecyl-3,3,3′,3′-teramethyl–indocarbocyanine perchlorate (DiI) as a lipophilic tracker and Hoechst 33,342 as a nucleic acid stain were obtained from Molecular Probes (Leiden, The Netherlands). Cholesterol (Chol) and N-succinimidyl S-acetylthioacetate (SATA) were purchased from Sigma (St. Louis, MO, USA). EGFP-S1 positive control DsiRNA (siRNA_GFP_) was purchased from Integrated DNA Technologies Inc. (Coralville, IA, USA). Other siRNA were obtained from Qiagen (Venlo, The Netherlands). Lipofectamine^TM^ 2000 was obtained from Invitrogen (Breda, The Netherlands). The hybridoma for producing E1/6-aa2 monoclonal antibody (mouse IgG1 against human VCAM-1) was kindly given by Dr. M. Gimbrone from the Harvard Medical School (Boston, MA, USA).

### 2.2. Preparation and Characterization of LNP

All LNP were prepared by ethanol injection using a NanoAssemblr™ Platform (NanoAssemblr™, Precision Nano-Systems Inc., Vancouver, BC, Canada). The lipid composition included DSPC, Chol, DMG-PEG, DSPE-PEG-Mal and cationic lipids (DOTAP or/and MC3) in a mol% ratio of 10:37.5:1.5:1:50. dLNP or mLNP refer to the LNP formulation containing 50 mol% of single cationic lipid, either DOTAP or MC3. dmLNP formulations contain three DOTAP/MC3 LNP, with different DOTAP-to-MC3 molar ratios of 5:45, 10:40 and 25:25, namely 5D/45M, 10D/40M and 25D/25M, as described in the text. When no ratio is indicated, the term dmLNP (or dmLNP_10D/40M_) specifically refers to the 10D/40M formulation. All LNP were prepared in the same way. Briefly, all lipids were dissolved in ethanol as an organic phase, with a final lipid concentration of 15 µM. Lipid mixtures were fluorescently labeled with the lipophilic dye DiI at 0.1 mol% of Total Lipids (TL), and siRNA were solubilized in 50 mM citrate buffer (pH 4). The aqueous phase and organic phase were loaded into two syringes that were connected to the inlet of a microfluidic cartridge. The siRNA: cationic lipid(s) ratio was 21 µg siRNA per µmol cationic lipid(s). The Flow Rate Ratio (FRR) was 3:1 and the Total Flow Rate (TFR) was 12 mL/min. After preparation of the LNP, ethanol was immediately removed by dialysis against HN buffer (10 mM N-2-hydroxyethylpiperazine-N′-2-ethanesulfonic acid (HEPES), 135 mM NaCl) pH = 6.9, overnight at 4 ℃. Antibodies were conjugated to the surface of LNP through coupling SATA-modified monoclonal anti-VCAM-1 antibodies with the maleimide group at the distal end of DSPE-PEG-Mal, as described before [28]. Coupled LNP were purified by ultracentrifugation and following dialysis against HN buffer (pH 7.4). All uncoupled LNP (uLN) and Ab_VCAM-1_ coupled LNP (AbLN) were characterized by measuring the phospholipid contents using a phosphorus assay to calculate TL concentration and measuring coupled antibodies by Peterson–Lowry assay using IgG as standard [29,30]. The encapsulated siRNA in LNP were measured by a fluorescence-based Quant-iT^TM^ RiboGreen kit (Invitrogen, Breda, The Netherlands) according to the manufacturer’s protocol. The siRNA encapsulation efficiency was calculated by taking the value of non-encapsulated siRNA in absence of 1% (*v*/*v*) Triton X-100 (Fi) divided by the total siRNA content in the presence of Triton X-100 (Ft): percentage of encapsulated siRNA = (1 − Fi/Ft) × 100%. Particle size and Polydispersity Index (PDI) were determined using a dynamic light scattering (DLS) in the volume weighing mode (NICOMP particle sizing systems, Santa Barbara, CA, USA) [31].

### 2.3. siRNA Integrity in Serum

To examine the integrity of encapsulated siRNA in different LNP formulations, LNP containing 130 ng of siRNA were incubated for 1.5 or 3 h at 37 ℃, in the absence or presence of 40% human serum. Naked siRNA sample at an equal amount as the encapsulated siRNA was used as control. After incubation, 2% (*v*/*v*) Triton X-100 and 1% (*v*/*v*) loading dye buffer (BioLabs, Leiden, The Netherlands) were added, and the samples were run in a 1% agarose gel at 120 V for 10 min. The bands were visualized via the ChemiDoc XRS system (Biorad, Veenendaal, The Netherlands).

### 2.4. Cell Culture and In Vitro Transfection

Human umbilical endothelial cells (HUVEC) were purchased from Lonza (Breda, The Netherlands) and cultured in EBM-2 medium supplemented with EGM-2 MV SingleQuot Kit Supplements and Growth Factors (Cat No. CC-3202, Lonza) at 37 ℃ in an incubator containing 5% CO_2_. All cell cultures were performed and maintained by the UMCG Endothelial Cell Facility. HUVEC from passages 5 to 7 were seeded on culture plates (Costar, Corning, USA) with various cell densities to obtain 50–90% confluency as indicated.

Where indicated, HUVEC were activated with 10 ng/mL of TNF-α (Boehringer, Ingelheim, Germany) for 4 h prior to adding LNP samples. uLN and AbLN containing RelA specific siRNA were added and incubated with the activated cells for 24 h at the indicated siRNA concentrations. TNF-α was present during the LNP incubation period. Subsequently, cells were washed and cultured for an additional 24 h in fresh medium without LNP and TNF-α. Additionally, at the 44 h time point, cells were re-challenged with LPS (1 μg/mL) for 4 h and next harvested for gene and/or protein expression analysis after being washed with phosphate-buffered saline (PBS). Lipofectamine-mediated siRNA_RelA_ transfection as control was also performed in the same activated HUVEC model, according to the manufacturer’s protocol.

### 2.5. Cell Association with LNP and siRNA Cargo Release in HUVEC

Endothelial cell association with LNP was analyzed by measuring the fluorescence of DiI-labeled LNP by flow cytometry. HUVEC were cultured in 12-well culture plates to 90% confluency. TNF-α (10 ng/μL) was added 2 h prior to the addition of 60 nmol TL/mL of DiI-labeled LNP. TNF-α activated and non-activated cells (resting cells) were incubated for 3 h with uLN or AbLN containing siRNA. After incubation, cells were washed with PBS twice and incubated with (0.025% *v*/*v*) trypsin-EDTA (Sigma, Ayrshire, UK). After 5 min incubation, FACS buffer (5% FBS in PBS) was added to stop trypsination, which was followed by harvesting cells into tubes. Flow cytometry analysis was performed using an ACEA NovoCyte™ flow cytometer (ACEA Biosciences, San Diego, CA, USA). To block VCAM-1 protein on EC, a 75-fold excess of anti-VCAM-1 antibodies was added to activated HUVEC prior to adding Ab_VCAM-1_ coupled LNP. All data were analyzed using the Kaluza Flow analysis software v 2.1 (Beckman Coulter, Brea, CA, USA).

SiRNA release from LNP in HUVEC was evaluated by determining the double fluorescent-labeled LNP system, where the lipids were labeled with DiI and the siRNA cargos were tagged with AlexaFluor_647_. (Activated) HUVEC seeded on a Lab-Tek^TM^ slide with 8 chambers (Nunc, Rochester, NY, USA) were incubated with LNP for 6 h at the lipid concentration of 60 nmol TL/mL. Hoechst 33,342 (20 µg/mL) was used for staining cell nuclei. Treated HUVEC were washed with serum-free medium twice and subjected to imaging within 1 h. Images were taken at excitation/emission wavelengths of 350/461 nm for Hoechst 33342, 550/570 nm for DiI and 651/672 nm for AlexaFluor_647_, using a Leica DM/RXA fluorescence microscope (Wetzlar, Germany) with a Quantimet HR600 image analysis software.

### 2.6. Gene Expression Analysis by qPCR

For gene expression analysis, after the transfection of TNF-α-activated HUVEC with different LNP formulations or lipofectamine, the total RNA of cells was isolated using TRIzol^®^ (Invitrogen, Cat No. 15596018) according to the protocol of the manufacturer. The concentration and purity of extracted RNA were measured using a NanoDrop^®^ ND-1000 UV-Vis spectrophotometer, and its integrity was analyzed by gel electrophoresis. Intact RNA samples with an expected OD260/OD280 ratio in the range of 1.8~2.0 were used for further processing. The cDNA synthesis and quantitative PCR (qPCR) were executed as described previously [32]. The Assay-on-Demand primers were purchased from Applied Biosystems (Nieuwekerk a/d Ijssel, The Netherlands), which were RelA (Hs00153294_m1), VCAM-1 (Hs00365486_m1), ICAM-1 (Hs00164932_m1), E-selectin (Hs00174057_m1), and housekeeping gene GAPDH (Hs99999905_m1). The qPCR reaction was performed with duplicates for each sample, and the threshold cycle values were calculated and averaged. Based on the comparative Ct method, all genes were normalized to the reference gene GAPDH to obtain the ΔCT value, and the relative mRNA level of a specific gene was calculated by 2^−ΔCT^. In addition, the final gene expression level of RelA was shown as mRNA fold change relative to the activated control group, which represents the HUVEC stimulated with TNF-a and LPS but without LNP treatment.

### 2.7. Protein Expression Analysis by Western Blot

For protein expression analysis, TNF-α-stimulated HUVEC were transfected with different LNP systems or lipofectamine associated with siRNA_RelA_ as described above and lysed in ice-cold RIPA buffer containing protease inhibitor and phosphatase inhibitor cocktail 1 (Sigma-Aldrich, Zwijndrecht, The Netherlands). Protein extracts (10 µg/lane) were separated on a 4-15% Mini-PROTEAN^®^ TGX™ precast gel (Bio-Rad, Cat no. 4561083) by electrophoresis and then transferred to a nitrocellulose membrane (Bio-Rad Laboratories, Utrecht, The Netherlands). The blots were blocked with 5% (*w*/*v*) nonfat dry milk (Campina, Friesland, The Netherlands) at room temperature for 1 h and subsequently incubated with different primary antibodies at 4 °C overnight. The primary antibodies included anti-RelA (Cat no. 8242, Cell Signaling Technology) diluted 1:2000 in 5% nonfat dry milk and anti-GAPDH (Cat no. sc-25778, Santa Cruz Biotechnology, Dallas, TX, USA) diluted 1:4000 in 5% nonfat dry milk. After washing with 0.1% (*v*/*v*) Tween-20 containing Tris-buffered saline (TBST), blots were incubated at room temperature for another hour with horseradish peroxidase (HRP)-conjugated goat anti-rabbit or anti-mouse secondary antibodies (diluted 1:5000 in 5% nonfat dry milk, Southern Biotech, Birmingham, AL, USA). After incubation with the immobilon Forte Western HRP substrate (Millipore, Billerica, MA, USA) for 5 min, membranes were analyzed using a GelDox XR system (Bio-Rad) with Image Lab software (Bio-Rad).

### 2.8. Cell Viability

The cell viability of EC transfected with different LNP/siRNA systems was evaluated using the Cell Counting Kit-8 (CCK-8) assay (HY-K0301, MedChemExpress, Monmouth Junction, NJ, USA). HUVEC were seeded in 96-well plates and activated with 10 ng/mL TNF-α for 4 h before the addition of lipofectamine or LNP containing 60 nM of control siRNA at 37 °C for 24 h, which was followed by 4 h incubation with 10% (*v*/*v*) CCK-8 solution in fresh complete medium. The absorbance at 450 nm was detected using a microplate reader. The absorbance of stimulated HUVEC without the addition of nanoparticles was considered as 100% in cell viability.

### 2.9. Zebrafish Husbandry and Injections

Zebrafish (*Danio rerio*) were bred and handled following the animal care guidelines and legislation of the European Convention for the Protection of Vertebrate Animals used for experimental and other scientific purposes. All the experiments complied with the rules and instructions of the local animal welfare committee of Leiden University. Most experiments were performed with the established transgenic zebrafish line Tg(*kdrl*:eGFP)^s8432^, expressing the green fluorescent-labeled protein GFP in EC [33]. A zebrafish model expressing GFP in macrophages named Tg(*mpeg1*:eGFP)^gl22^ was used to demonstrate the macrophage uptake of nanoparticles [34]. Fertilization was achieved by natural spawning in the morning, and then, eggs were collected and raised at 28.5 °C in egg water with 60 μg/mL instant ocean sea salts. Embryos were anesthetized with a 0.01% tricaine buffer and embedded in 0.4% agarose with tricaine before microinjection. LNP solutions were loaded into borosilicate needles previously prepared using a micropipette puller (P-97, Sutter instrument, Co., Ltd., Novato, CA, USA). All LNP formulations containing AlexaFluor_647_ siRNA (1 nL) or siRNA_GFP_ (2 nL) were injected into 54 h post-fertilization (hpf) embryos through duct of Cuvier, as described previously [35]. Fluorescence microscopy was used to examine the results of injection. Successfully injected embryos displayed circulating LNP, and the injection did not cause damage to the yolk.

### 2.10. Zebrafish Imaging and Quantification of GFP Silencing in Zebrafish Embryos

For visualization of the in vivo distribution and behavior of siRNA-containing LNP, at least five zebrafish embryos were randomly selected from 20~40 successfully injected embryos for imaging using an SP8 confocal microscope (Leica Application Suite X software, version 3.5.5.19976, Leica Microsystems, Wetzlar, Germany). Confocal z-stacks were captured using a ×10 air objective (HCX PL FLUOTAR) and a ×40 water-immersion objective (HCX APO L). For whole-embryo pictures, 3~5 overlapping z-stacks were captured to cover the complete embryo. Laser intensity, gain and offset settings were identical between stacks and experiments. For the in vivo visualization and quantification of GFP silencing, seven embryos were imaged at indicated hours post-injection (hpi) under the same conditions as mentioned above. Quantification was performed on ×40 confocal z-stacks using methods previously described [36]. The images were processed, and the total fluorescence of the whole image was calculated using Fiji ImageJ 1.53q_Java 1.8.0_172.

### 2.11. Statistical Analysis

The data were statistically analyzed by Student’s *t*-test for two-group comparisons or one-way ANOVA followed by Bonferroni post hoc correction or Tukey’s test for multigroup comparisons. The statistical analyses were performed using GraphPad Prism Software v. 8.3.0 (GraphPad Prism Software, San Diego, CA, USA). Differences were considered significant if *p* < 0.05.

## 3. Results

### 3.1. Preparation and Characterization of LNP with Different Cationic Lipid Compositions

Three types of LNP were formulated by varying the composition of cationic lipids, namely dLNP, mLNP and dmLNP, with the dmLNP here containing 10 mol% of DOTAP and 40 mol% of MC3. The cationic lipids made up 50 mol% of TL in all LNP formulations, and other lipid components were identical in each formulation. In addition, since vascular cell adhesion protein 1 (VCAM-1) is strongly upregulated by activated EC, endothelial cell-targeted LNP were also prepared by conjugating antibodies against VCAM-1 and used in some experiments as indicated (Figure 1A). The physicochemical characteristics of these LNP were given in Table 1. uLN had particle diameters of 59–88 nm, with the dLNP being most homogeneous (PDI ≤ 0.2). All LNP formulations had zeta potentials between −0.44 and −6.35 mV (data not shown), with siRNA encapsulation efficiencies of >90%. The three LNP formulations preserved the integrity of siRNA after 3 h of incubation with serum, while the naked siRNA were completely degraded (Figure 1B). Coupling antibodies to the LNP surface increased particle sizes to 110–139 nm without significant changes in other parameters. Additionally, AbLN also showed similar siRNA protective ability as their uLN analogues to resist siRNA degradation in serum, even after 3 weeks of storage (Appendix A).

### 3.2. Influence of the Cationic Lipid Composition on LNP-Cell Association and Gene Silencing in HUVEC

How the cationic lipid composition of LNP formulations influences endothelial association and siRNA transfection in EC is still unclear. Thus, the endothelial association and transfection efficiencies of three LNP formulations with different cationic lipid compositions and their antibody conjugates (uLN and AbLN) were investigated in TNF-α-stimulated HUVEC. As shown in Figure 2A, dLNP showed the highest association with HUVEC, which was followed by dmLNP, and mLNP showed the lowest association. Although mLNP showed little association with activated HUVEC, surface modification with anti-VCAM-1 antibodies significantly increased endothelial cell-association, which was also the case for the dmLNP formulation. To further identify the association specificity of different VCAM-1 targeted LNP with activated EC, a 75-fold of excess anti-VCAM-1 antibodies was added to compete for antibody-mediated specific association of AbLN. A significant decrease in LNP-endothelial association was observed in the Ab-mLNP and Ab-dmLNP groups, while no significant decrease was measured in the Ab-dLNP group. This suggests that endothelial cell association with the Ab-dLNP formulation is independent of antibody binding. In terms of gene silencing, only MC3-containing LNP formulations (mLNP and dmLNP) were able to silence the expression of RelA, with the dmLNP showing a superior down-regulation of RelA mRNA expression. In addition, at the tested siRNA concentration, a significant difference in knockdown between targeted and non-targeted LNP was only shown in the mLNP formulation, with more knockdown occurred in the Ab-mLNP group.

### 3.3. Influence of the DOTAP/MC3 Ratios in the dmLNP Formulations on Cell Association and Gene Silencing in EC

We next evaluated how the ratio of the permanent cationic lipid to the ionizable cationic lipid influenced the endothelial association and transfection ability of dmLNP in EC. Three dmLNP (5D/45M, 10D/40M and 25D/25M) were incubated with TNF-α-activated HUVEC, respectively. As shown in Figure 3A, the cell association capability of dmLNP depends on the ratio of DOTAP to MC3. Higher DOTAP content in the formulation resulted in higher endothelial cell association, and conjugation with anti-VCAM-1 antibodies further increased LNP association with activated EC. To investigate gene silencing, activated HUVEC were transfected with different LNP at siRNA concentrations of 15 and 60 nM (Figure 3B). The formulations including higher MC3 contents (10D/40M and 25D/25M) exhibited a dose-dependent reduction in RelA mRNA expression. 10D/40M possess higher gene silencing capability, as demonstrated by the minimum effective siRNA concentration (15 nM) shown in the 10D/40M formulation compared to other formulations. All Ab_VCAM-1_-coupled forms showed more RelA down-regulation at both siRNA concentrations compared to their uncoupled counterparts. Because of its good transfection ability, the 10D/40M formulation was chosen for further studies.

### 3.4. Intracellular Release of Encapsulated siRNA from LNP in EC

The efficient and timely intracellular release of siRNA from LNP is one of the requirements for achieving efficient siRNA-mediated gene silencing. Here, we investigated the release of siRNA from dLNP, mLNP and dmLNP by tracking the fluorescence of the lipid carrier (DiI, red) and the encapsulated siRNA (Alexa_647_-siRNA, green). As shown in Figure 4, siRNA were hardly dissociated from dLNP, as evidenced by the co-localization of the green fluorescence of siRNA and the red fluorescence of DiI-labeled LNP. Incorporating MC3 in the formulation resulted in a release of the encapsulated siRNA from dmLNP into the cytoplasm, as evidenced by the appearance of green fluorescence in the merged image. Much less fluorescence of DiI-labeled LNP and Alexa_647_-siRNA was observed in the mLNP group compared to that of the dLNP and dmLNP groups, indicating the limited cellular internalization of mLNP in EC. Its Ab_VCAM-1_ coupled form showed an increased cellular uptake of particles as well as a subsequent siRNA intracellular release (Appendix A). Combining all data, we conclude that MC3 lipid is the key lipid in LNP formulations to achieve an intracellular release of siRNA. In combination with DOTAP, the dmLNP formulation achieved both improved cellular uptake and siRNA intracellular release, thus enhancing siRNA transfection in EC.

### 3.5. Anti-Inflammatory Potential of LNP/siRNA_RelA_ in Inflammatory-Activated EC In Vitro

The siRNA_RelA_-mediated gene silencing of RelA expression blocks the NF-κB signaling pathway, as evidenced by the reduced expression of downstream inflammation-associated genes, such as VCAM-1, ICAM-1, and E-selectin [31]. To determine whether Ab_VCAM-1_-coupled LNP/siRNA_RelA_ systems have the potential to attenuate endothelial inflammatory activation, we activated HUVEC with TNF-α and re-challenged the cells with LPS after incubation with LNP. Lipofectamine-assisted siRNA_RelA_ transfection was applied as the positive control, while activation only without siRNA transfection was taken as the negative control for the knockdown experiment. A significant reduction in RelA expression at mRNA and protein levels could be observed in the Ab-mLNP and Ab-dmLNP treated groups, while knockdown was absent in the Ab-dLNP treated group (Figure 5A). The RelA knockdown in Ab-dmLNP-treated cells was comparable to that in the lipofectamine-treated group. The similar down-regulation trends of VCAM-1, ICAM-1 and E-selectin mRNA expression were observed in the Ab-mLNP and Ab-dmLNP groups (Figure 5B). The cell viability of LNP-treated EC was significantly higher than that of lipofectamine-treated cells (Appendix A). Compared with lipofectamine, the dmLNP formulation is a less toxic siRNA delivery system with comparable transfection capability as well as anti-inflammatory effects on inflammatory-activated EC.

### 3.6. In Vivo LNP/siRNA Behavior in Zebrafish Embryos

The *kdrl*: eGFP transgenic zebrafish model that expresses eGFP protein in EC was used to track the nanoparticle behavior of LNP in vivo (Figure 6A). Here, we first studied the biodistribution of Dil-labeled LNP at the tissue level (Figure 6B). In vivo, the three LNP were associated with the blood vasculature at indicated time points but with different distribution patterns. dLNP associated with EC and were cleared from the circulation at 1 hpi, as demonstrated by the absence of the white flow of particle signals in the circulation. The biodistribution pattern of dLNP did not change during the entire measurement period, and dLNP mainly associated with scavenger EC (sEC) in caudal hematopoietic tissues (CHT) and caudal veins [37]. However, mLNP circulated freely within the lumen of various blood vessels at 1 hpi. At later time points, mLNP showed a more clearly visible association with CHT-EC and eventually were removed from the circulation at 48 hpi. dmLNP showed a distribution pattern of circulating LNP (similar to mLNP) and endothelial-associated LNP (similar to dLNP) at 1 hpi. At later time points, fewer dmLNP circulated and more associated with CHT-EC. dmLNP exhibited increased endothelial association compared to mLNP. The biodistribution patterns of all LNP throughout the embryos were shown in Appendix A, showing that the dLNP and dmLNP formulations displayed particle retention in the injection area around yolk at 24 hpi. The phenomenon of vascular extravasation (leakage through blood vessels) can also be observed in mLNP and dmLNP (Figure 6, white rectangles), with the dmLNP showing a weaker signal compared to mLNP. Additionally, we found that MC3-containing LNP samples not only associated with EC but also with other cells after 12 hpi (Figure 6B, red triangles), which were most likely macrophages. Therefore, we further studied the interaction between mLNP and macrophages in a Tg (*mpeg*: eGFP) zebrafish model, as shown in Appendix A. The co-localization of mLNP and macrophages demonstrated the phagocytosis of mLNP by macrophages.

The siRNA release from LNP was also visualized by analyzing the signal of fluorescently-labeled siRNA, and the magenta fluorescent indicated the co-localization of LNP and siRNA (Figure 6A,C). At 1 hpi, all three LNP co-localized with siRNA in the vasculature and EC. The release of siRNA was observed at 3 hpi, as demonstrated by the blue fluorescence, but the majority of siRNA associated with LNP. At later time points, more siRNA were released from LNP. After 12 hpi, a difference in siRNA biodistribution of the three LNP was observed. In the mLNP and dmLNP groups, siRNA were observed throughout the vasculature, indicating that most siRNA were released and diffused. However, in the dLNP group, the siRNA fluorescence was restricted to the endothelial region where dLNP were immobilized. For all LNP formulations tested, siRNA associated with LNP were still visible at 48 hpi, which was confirmed by detailed single siRNA channel images (Appendix A). SiRNA release from mLNP in macrophages was also observed (Appendix A). The siRNA fluorescence was evident in macrophages at 1, 3 and 12 hpi but barely found at 24 hpi, indicating that a quicker degradation of siRNA occurred after being trapped in macrophages compared to in sEC. In summary, dmLNP possess hybrid characteristics with mLNP and dLNP, showing enhanced EC-association and more siRNA release in vivo.

The Ab_VCAM-1_ coupled mLNP formulation is a promising formulation for the specific targeting of diseased EC, as demonstrated by the high specificity of targeting to activated EC in vitro. The specific targeting of LNP could not be assessed in vivo in zebrafish. However, in a proof-of-concept study, we found that the biodistribution of nanoparticles and siRNA cargos in the Ab-mLNP formulation did not change compared to its uncoupled LNP, which implies that the antibody modification strategy per se does not affect LNP biodistribution and cargo release from mLNP (Appendix A).

### 3.7. Gene Silencing Potency of MC3 Based LNP Loaded with siRNA against GFP in Zebrafish

In the same zebrafish model, we next investigated whether mLNP and dmLNP are able to transfect EC in vivo. LNP containing siRNA against GFP (siRNA_GFP_) or control siRNA (siRNA_ctrl_) were injected into zebrafish embryos, and untreated embryos were used as control. At 24 hpi, GFP expression was visualized and shown in tissue-level images (Figure 7A–C). Reduced GFP fluorescence was only observed in the dmLNP/siRNA_GFP_-treated group. The GFP knockdown was quantified as mean fluorescence intensity (MFI) by ImageJ (Figure 7D). GFP fluorescence was reduced to 43.8% in the dmLNP/siRNA_GFP_ group compared to that in the dmLNP/siRNA_ctrl_ group. These results confirm that the dmLNP formulation can be used for siRNA transfection in EC in vivo to significantly down-regulate the target genes. mLNP were able to silence the target genes in vitro but did not show significant gene silencing in vivo.

## 4. Discussion

EC are important therapeutic targets due to their role in the pathology of many diseases. However, in general, EC are relatively refractory to carrier-assisted transfection. In the current work, we described the preparation and characterization of LNP/siRNA systems based on different cationic lipid compositions, and we also comparatively evaluated their performances for the delivery of siRNA into EC both in vitro and in vivo. All LNP formulations showed comparable physicochemical properties, including nanoscale in size (<200 nm), high siRNA encapsulation efficiency (>90%), nearly neutral surface charge and siRNA protective ability. The results indicate that the physicochemical properties of our LNP are independent of the cationic lipid composition. This is not completely consistent with some studies conducted with a series of LNP formulations with various cationic lipid compositions, where it was concluded that physicochemical characteristics depend on the selection of cationic lipid [38,39]. The optimal physicochemical properties of siRNA carriers are important prerequisites for successful siRNA delivery [40]. All LNP formulations tested have the potential to successfully deliver siRNA into EC, which complies with further developmental perspectives.

Basha et al. have verified that the cationic lipid composition determines gene silencing profile by influencing cellular and intracellular siRNA delivery routes [41]. For example, a higher LNP uptake of DinKC2-DMA LNP into antigen-presenting cells showed better gene silencing potency than other formulations [42]. Another study has proven that some cationic LNP formulations were superior to ionizable LNP for in vitro transfection, which is likely a result of higher cellular uptake, which was attributed to permanently charged cationic lipids [38]. Our in vitro studies demonstrate that when ionizable cationic lipid-MC3 is used, limited cell association with EC is a rate-limiting step for transfection. When permanent cationic lipid-DOTAP is applied, the poor transfection efficiency of dLNP is likely attributed to the inefficient intracellular release of encapsulated siRNA into the cytoplasm. This finding is in agreement with another study reporting the poor transfection efficiency of DOTAP-containing lipoplexes, which corroborated to inefficient DNA release from DOTAP cationic vesicles into the cytoplasm [43]. Yung et al. reported a combination formulation with amine and tertiary amine cationic lipids, which achieved superior delivery of microRNA-21 [44]. In line with this study, we found that the combination strategy of using permanent and ionizable cationic lipids in one LNP formulation is capable of enhancing siRNA transfection due to the combined effects of enhanced endothelial cell association and efficient intracellular release of siRNA. Endothelial association depends on the permanent cationic lipid content, showing that more DOTAP included results in more LNP-endothelial association. However, it does not mean that more endothelial association achieves higher transfection efficacy. Most gene silencing was observed in the 10D/40M LNP formulation, which suggests a balance between endothelial association and efficient intracellular siRNA release. Lipofectamine is a widely used transfection agent with high transfection efficiency in vitro but also with a high cytotoxicity profile [45]. Compared to lipofectamine, the dmLNP formulation showed less cytotoxicity in HUVEC but comparable transfection efficiency and anti-inflammatory effects. In addition, in zebrafish, no adverse effects of LNP were observed, implying therapeutic potential for further in vivo applications.

The rational design of advanced carrier-mediated therapeutic strategies must be preceded by a comprehensive understanding of primary relevant nano-biointeractions, and the embryonic zebrafish is a good model to qualitatively predict in vivo nanoparticle interactions with specific cells [36,46,47]. As a screening or/and optimization platform, the zebrafish model allows the real-time assessment of the in vivo fate of LNP in a precise and flexible way. A recent study revealed a charge-dependent mechanism underpinning LNP recognition, uptake and processing in a zebrafish model for anionic LNP that preferentially targeted hepatic reticuloendothelial system (RES) cell types [48]. Another study demonstrated that the in vivo biodistribution is determined by the surface charge of liposomes [36]. Interestingly, in our study, we found that a charge-independent mechanism influences the selective accumulation of LNP within blood vessels, especially sEC. The in vivo biodistribution of LNP was determined by the cationic lipid composition rather than the surface charge of LNP. More DOTAP contributes to greater endothelial association, while more ionizable MC3 allows longer circulation. This indicates the importance of considering not only the physicochemical properties of the formulated lipid carriers but also the intrinsic properties of lipids itself. In terms of in vivo gene silencing in EC, we found that the dmLNP reduced GFP fluorescence expression in vivo while the mLNP did not, which might be due to the lower endothelial association compared to the dmLNP formulation. In this paper, we showed that the zebrafish model is a useful platform for tracking the delivery of nanoparticles and evaluating nanoparticle efficacy in vivo.

In summary, we developed an efficient and low-cytotoxic lipid nanoparticle-based siRNA delivery system with a combination of a permanent cationic lipid-DOTAP and an ionizable cationic lipid-MC3. In addition, we also show a relation between specific delivery routes and high transfection efficiency by comparing the nanoparticle behavior and transfection performances of single cationic lipid-based formulations and the DOTAP/MC3 formulation. Further in vivo toxicity, pharmacokinetic and pharmacological studies are needed to evaluate the dmLNP system for therapeutic applications and translation to the clinic.

## Figures and Tables

**Figure 1 pharmaceutics-14-02086-f001:**
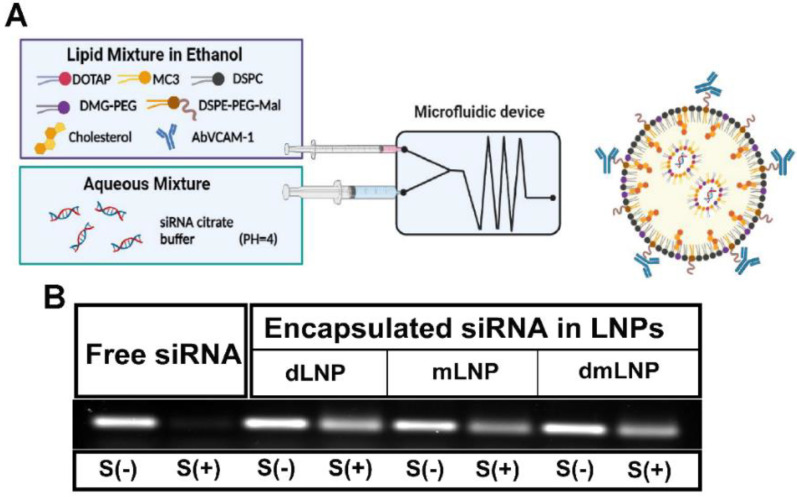
Preparation and characterization of LNP with different cationic lipid compositions. (**A**) Schematic representation of LNP preparation. (**B**) The agarose gel electrophoresis image of non-encapsulated siRNA and siRNA encapsulated in LNP. Intact free siRNA and three of LNP/siRNA samples were incubated with (+) or without (−) 40% serum (S) for 3 h at 37 ℃.

**Figure 2 pharmaceutics-14-02086-f002:**
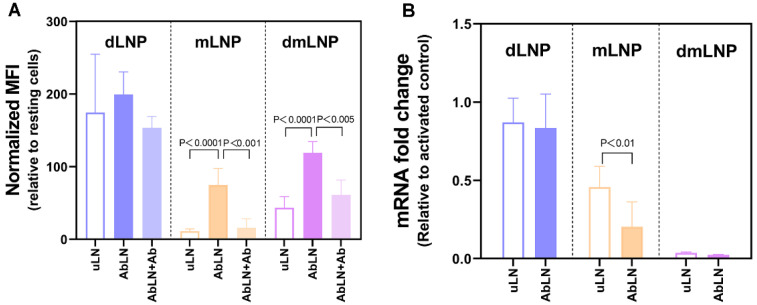
Endothelial association and transfection activities of three LNP formulations with different cationic lipid compositions. (**A**) Specificity and extent of cell association of LNP to HUVEC, as determined by flow cytometry. TNF-α-activated HUVEC were incubated with uLN or AbLN of three formulations for 3 h. The specificity of cell association to VCAM-1 was investigated by co-incubation with an excess of anti-VCAM-1 antibodies together with AbLN. Data are presented as normalized mean fluorescence intensity (MFI) ± SD values of three independent experiments. HUVEC_resting_/HUVEC_activated only_ = 1. (**B**) Transfection activities of uncoupled and antibody-coupled LNP formulations loaded with siRNA_RelA_ in activated EC. TNF-α-activated HUVEC were transfected with uLN or AbLN containing siRNA_RelA_ at 100 nM concentration for 48 h and next processed for qPCR as described in the Section 2. Data are shown as relative fold change in gene expression ± SD of 3 independent experiments.

**Figure 3 pharmaceutics-14-02086-f003:**
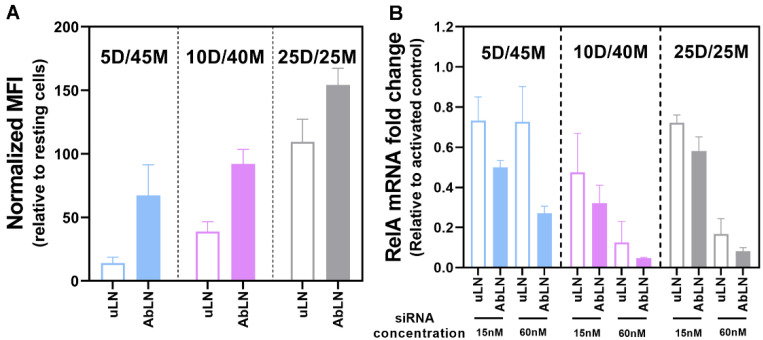
Effect of the DOTAP/MC3 ratios in dmLNP formulations on cell association and transfection in activated HUVEC. (**A**) Endothelial association of dmLNP formulations with increasing DOTAP/MC3 ratios (5D/45M, 10D/40M and 25D/25M). TNF-α activated HUVEC were incubated with LNP for 3 h, and EC-associated-DiI fluorescence was determined by flow cytometry. Data are normalized MFI ± SD of three independent experiments. (**B**) SiRNA-based gene silencing in activated EC mediated by the LNP formulations with different DOTAP/MC3 ratios, at the siRNA concentration of 15 nM or 60 nM. Data are from 3 independent experiments and shown as mRNA fold change ± SD.

**Figure 4 pharmaceutics-14-02086-f004:**
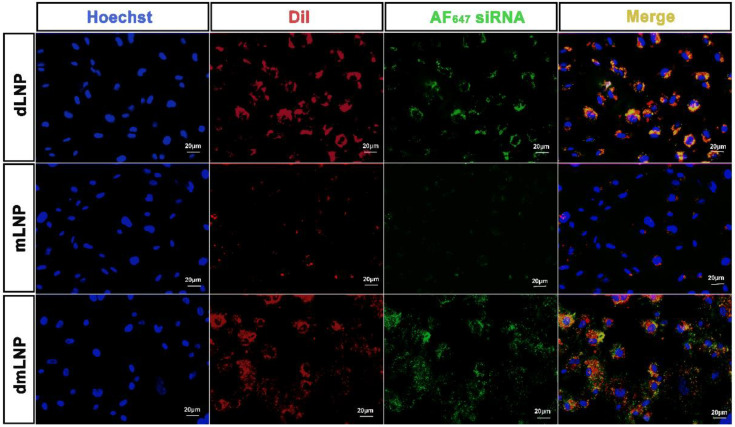
Intracellular release of siRNA from dLNP, mLNP and dmLNP (10D/40M) in EC. HUVEC were incubated for 6 h with three LNP formulations labeled with DiI (red) and loaded with AlexaFluo_647_-siRNA (green), the nuclei were stained using Hoechst 33,342 (blue). Presented datasets are representative fluorescence microscopy images of 2 independent experiments. Scale bar: 20 µm.

**Figure 5 pharmaceutics-14-02086-f005:**
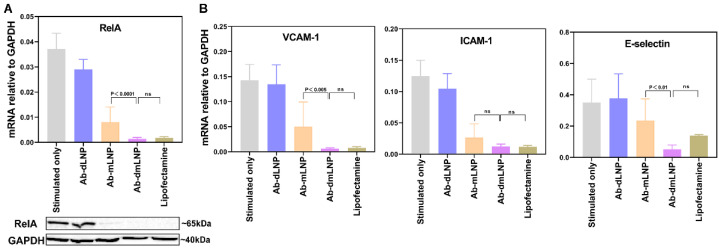
Ab-dmLNP/siRNA_RelA_ attenuates endothelial inflammatory activation after down-regulation of NF-κB P65 (RelA), which is as effective as lipofectamine. (**A**) The mRNA and protein levels of RelA after siRNA_RelA_-based transfection mediated by different LNP systems and lipofectamine. (**B**) Gene silencing of inflammation-related genes VCAM-1, ICAM-1 and E-selectin, after LNP or lipofectamine-mediated RelA knockdown. Data are represented as mean values ± SD of 3 independent experiments.

**Figure 6 pharmaceutics-14-02086-f006:**
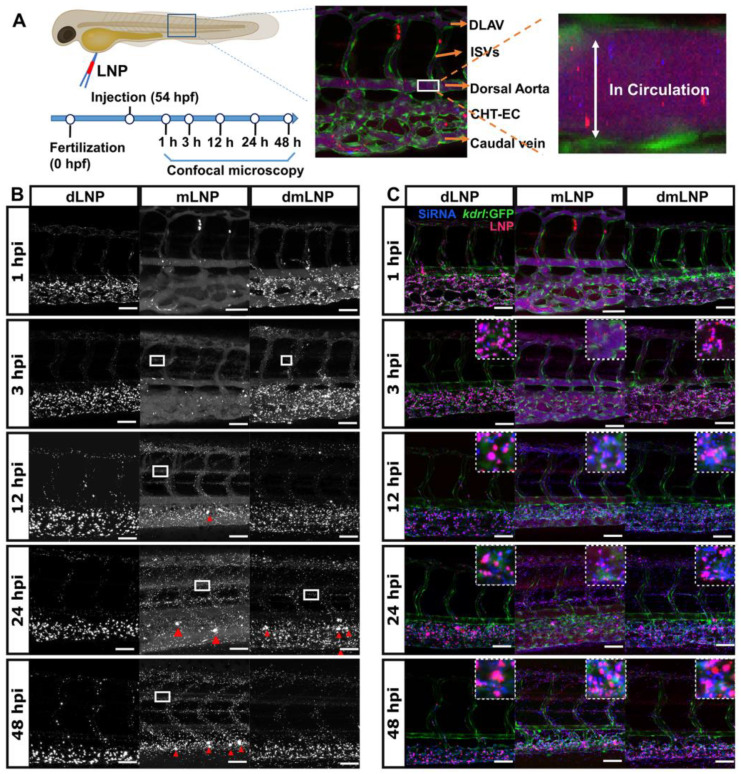
Biodistribution of LNP and siRNA in zebrafish embryos at tissue-level view. (**A**) Schematic diagram of zebrafish microinjection. First, 1 nL of Dil-labeled LNP was injected into the duct of Cuvier of 54 hpf zebrafish embryos, and confocal images were taken at 1, 3, 12, 24 and 48 h after injection. Tissue-level images of LNP distribution in Tg (*kdrl*:eGFP) embryos were shown with the white boxes indicating LNP in the circulation. CHT-EC: caudal hematopoietic tissue endothelial cells, DLAV: dorsal longitudinal anastomotic vessel. ISV: intersegmental vessel. (**B**) The bright-field images of LNP biodistribution at multiple hpi (s). Macrophage-uptake of LNP is marked with red triangles, and extravasation of LNP is indicated by white rectangles. (**C**) The fluorescence images of the biodistribution of LNP (red) and siRNA (blue) at different time points. The zoomed-in images are displayed in the white squares. Scale bar of (B,C): 200 µm.

**Figure 7 pharmaceutics-14-02086-f007:**
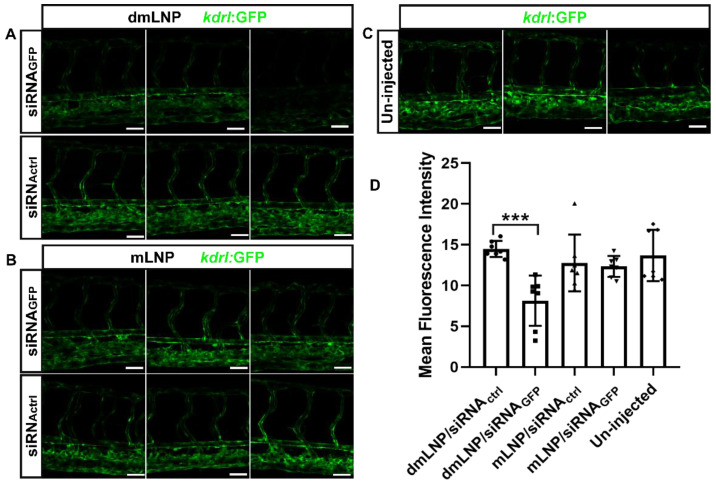
Effective in vivo GFP silencing mediated by the DOTAP/MC3 LNP/siRNA system (10D/40M) in Tg (*kdrl*: eGFP) zebrafish. Confocal microscopy images of GFP (scale bar: 200 µm) in (**A**) dmLNP containing siRNA_GFP_ or siRNA_ctrl_ (dmLNP/siRNA_GFP_ vs. dmLNP/siRNA_ctrl_); (**B**) mLNP containing siRNA_GFP_ or siRNA_ctrl_ (mLNP/siRNA_GFP_ vs. mLNP/siRNA_ctrl_); (**C**) Untreated group. (**D**) Quantification of GFP fluorescence in entire images by ImageJ (*n* = 7). The data represent the mean ± SD. *** *p* < 0.001according to one-way ANOVA, Tukey’s multiple comparisons test. Scale bar of (**A**–**C**): 200 µm.

**Table 1 pharmaceutics-14-02086-t001:** Physicochemical properties of siRNA-containing LNP formulations based on different cationic lipid compositions.

Category	Sample	Size (nm)	Polydispersity Index (PDI)	siRNA Encapsulation Efficiency (EE, %)	Ab Conjugated/LNP (µg/µmol TL)
uLN	dLNP	88 ± 16	0.11 ± 0.05	98 ± 1	-
	mLNP	59 ± 7	0.21 ± 0.09	91 ± 8	-
	dmLNP	63 ± 9	0.28 ± 0.11	97 ± 1	-
AbLN	Ab-dLNP	139 ± 25	0.22 ± 0.05	98 ± 1	100 ± 19
	Ab-mLNP	118 ± 10	0.34 ± 0.06	91 ± 2	85 ± 15
	Ab-dmLNP	110 ± 6	0.34 ± 0.09	90 ± 7	92 ± 23

Data are presented as means of 4–5 preparations ± SD. dmLNP here refers to the 10D/40M formulation.

## Data Availability

Data are contained within this article.

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
