# Peer review of "Development of a Combined Lipid-Based Nanoparticle Formulation for Enhanced siRNA Delivery to Vascular Endothelial Cells"

_pharmaceutics, 2022, doi:10.3390/pharmaceutics14102086_

Round 1

Reviewer 1 Report

The authors describe the development of a novel LNP formulation for siRNA delivery. Interestingly, the authors studied the combination of well-known cationic and ionizable lipids for the generation of LNPs with improved siRNA transfection efficiency. The manuscript reflects a well-performed and structured work containing detailed physicochemical and biological analyses of the LNP formulation. However, preliminary experiments in mice would have been ideal to be tested. 

-       Cationic lipids are frequently associated with cytotoxicity. On the other hand, MC3 lipids have been also associated with toxicity in vivo due to their poor biodegradability rates. However, the authors did not include any description of these negative effects. Please, comment about it. 

-       It is not quite clear whether the dmLNP formulation is a mixture of dLNP and mLNP, or the lipids were mixed and then dmLNPs were synthesized. Please, include a more detailed explanation. 

-       Figure 4 should be optimized; colors are oversaturated and no scale bar was included. 

Reviewer 2 Report

This manuscript evaluated the siRNA delivery efficiency of antibody targeted Lipid-based nanoparticle to endothelial cells. The AbVCAM-1 coupled LNP composed of permanent cationic DOTAP and ionisable MC3 and/or their combinations. The effectiveness of siRNA release from various LNPs and their capacity for gene silencing were investigated. They also looked into the potential of AbVCAM-1-coupled LNP/siRNA to attenuate endothelial inflammatory activation. The in vivo biodistribution kenetics of dLNP, mLNP and dmLNP were compared Zebrafish model and GFP knockdown were evaluated as well. The information provided from this study provide valuable knowledge of advanced LNP formulation for siRNA delivery to endothelia cells by potential combination of DOTAP and MC3. The study was well designed and fit well into the scope of journal.

1.      How ee% of siRNA is calculated?

2.      Title of Y-axis of Figure3B confuse. Maybe use “RelA mRNA fold change”? in line 308-309, I think author want to say “..a SIGNIFICANT difference in knockdown between targeted and non-targeted LNP was only shown ….”.

3.      the Statement on lines 332-337, “The formulations including higher MC3 contents … to their uncoupled counterparts”. Is there any stats on these comparison?

4.      It is interesting to see that the strongest attenuation of endothelial inflammation occurred in both Ab-dmLNP vs. lipofectamin (figure5). Lipofectamin is another   kind of cationic-lipid transfection agent, similar to DOTAP, what is the possible reason that lipofectamine was the most effective in down-regulation of mRNA expressions of VCAM-1, ICAM-1 and E-selectin? Whereas dLNP that contained DOTAP has the least RelA mRNA fold changes?

5. Please check reference style, it needs to meet journal guideline 

Reviewer 3 Report

The manuscript generally well written and designed 

Some minor corrections are required

1. Rewrite the abstract including more results as it appear as abstract for review article

2. In methodology try to describe every abbreviation when mentioned first time

3. Add more discussion for the results as discussion appear as a redundant for the results
